# NMR Studies of Two Lysine Based Dendrimers with Insertion of Similar Histidine-Arginine and Arginine-Histidine Spacers Having Different Properties for Application in Drug Delivery

**DOI:** 10.3390/ijms24020949

**Published:** 2023-01-04

**Authors:** Nadezhda N. Sheveleva, Irina I. Tarasenko, Mikhail A. Vovk, Mariya E. Mikhailova, Igor M. Neelov, Denis A. Markelov

**Affiliations:** 1Saint Petersburg State University, 7/9 Universitetskaya Nab, 199034 Saint Petersburg, Russia; 2Institute of Macromolecular Compounds, Russian Academy of Sciences, Bolshoi Prospect 31, V.O., 199004 Saint Petersburg, Russia; 3School of Computer Technologies and Control, Saint Petersburg National Research University of Information Technologies, Mechanics and Optics (ITMO University), Kronverkskiy Pr. 49, 197101 Saint Petersburg, Russia

**Keywords:** peptide dendrimer, NMR spectroscopy, NMR relaxation, histidine, arginine, pairing effect

## Abstract

In this paper we study two lysine-based peptide dendrimers with Lys-His-Arg and Lys-Arg-His repeating units and terminal lysine groups. Combination of His and Arg properties in a dendrimer could be important for biomedical applications, especially for prevention of dendrimer aggregation and for penetration of dendrimers through various cell membranes. We describe the synthesis of these dendrimers and the confirmation of their structure using 1D and 2D Nuclear Magnetic Resonance (NMR) spectroscopy. NMR spectroscopy and relaxation are used to study the structural and dynamic properties of these macromolecules and to compare them with properties of previously studied dendrimers with Lys-2Arg and Lys-2His repeating units. Our results demonstrate that both Lys-His-Arg and Lys-Arg-His dendrimers have pH sensitive conformation and dynamics. However, properties of Lys-His-Arg at normal pH are more similar to those of the more hydrophobic Lys-2His dendrimer, which has tendency towards aggregation, while the Lys-Arg-His dendrimer is more hydrophilic. Thus, the conformation with the same amino acid composition of Lys-His-Arg is more pH sensitive than Lys-Arg-His, while the presence of Arg groups undoubtedly increases its hydrophilicity compared to Lys-2His. Hence, the Lys-His-Arg dendrimer could be a more suitable (in comparison with Lys-2His and Lys-Arg-His) candidate as a pH sensitive nanocontainer for drug delivery.

## 1. Introduction

Dendrimers are regular hyperbranched monodisperse macromolecules with a well-defined spherical structure. These nanosized macromolecules are widely used in biomedicine [1,2,3,4,5,6] as carriers in gene and drug delivery [7,8,9,10,11,12,13,14]. In the latter case, dendrimers contribute to an increase in the solubility of drugs, a decrease in their toxicity for normal cells, and a prolongation in their circulation time in blood flow [15,16,17]. The development of drugs and gene delivery carriers based on lysine and peptide dendrimers has been described in many studies [18,19,20,21,22]. The usage of these dendrimer carriers could provide a synergistic effect due to the bioactive action of the dendrimers themselves, which have antimicrobial [23,24,25,26,27,28] and antiangiogenic activity [29,30,31,32].

The search for new drugs and methods for their safe delivery, as well as the encouraging results of cytological studies of dendrimers [22,31,32,33,34,35,36,37], inspire researchers to develop new dendrimer macromolecules [38,39]. The possibility of step-by-step synthesis allows control over their structure and composition, as well as permitting the introduction of functional groups into the core, inner, or terminal segments. This functionalization is a reliable way to obtain dendrimers with tailored characteristics for various purposes [40,41,42]. Thus, an overall positive charge of the dendrimer macromolecules is an important characteristic, which is required for effective cellular uptake. The lysine-based dendrimers functionalized with additional lysine and glycine amino acid residues suppressed the growth of cancer cells [22]. Lysine dendrimers with double arginine insertions into spacers between neighboring lysine branching points were effective as siRNA carriers [35]. The introduction of histidine amino acid residues into the structure gives new properties [43] to the dendrimer. One of them is the ability to be deuterated. The proton at the C_2_ atom of the histidine imidazole ring is replaced by deuterium upon heating [44,45,46]. It showed the possibility of controlled deuterium labeling of the Lys-2His dendrimers. In addition, the chemical structure of these dendrimers remains stable after deuteration [47]. Another important effect is the change of the global conformation of histidine-containing dendrimer with the changing of the pH level. We have found that the Lys-2His dendrimer undergoes a conformational transition from a swollen conformation at low pH to a collapsed conformation at normal pH [48]. This circumstance can play an important role for using the histidine-containing dendrimer as a nanocontainer.

The above-mentioned results confirm that peptide dendrimers modified with various amino acid residues have great potential for biomedical applications. In particular, a change in the size of the Lys-2His dendrimer in the collapsed conformation can lead to the aggregation of dendrimer macromolecules, which, of course, can adversely affect their biomedical application. That is why this article is devoted to the synthesis and study of new Lys-His-Arg and Lys-Arg-His dendrimers with an intermediate chemical structure between the Lys-2His dendrimer (having a collapsed conformation at normal pH) and the Lys-2Arg dendrimer (having a swollen conformation at normal pH). In contrast to the lysine and lysine-based peptide dendrimers synthesized and studied by us earlier [47,48,49,50,51,52,53,54,55,56,57,58], the considered dendrimers contain two different types of amino acid residues between the branching points: histidine and arginine. The only difference between the two dendrimers is the order in which the histidine and arginine residues are inserted into spacers between lysine branching points. NMR spectroscopy and relaxation methods were used for their investigations. In this work, it has been shown that this method is sensitive to the collapsed conformation of the peptide dendrimer.

## 2. Results

### 2.1. Analysis of ^1^H and ^13^C NMR Spectra

Structural analysis of the newly synthesized second-generation dendrimers Lys-Arg-His and Lys-His-Arg was carried out using the methods of one-dimensional and two-dimensional NMR spectroscopy. Figure 1 and Figure 2 show the ^1^H and ^13^C NMR spectra of the Lys-Arg-His dendrimer. Figure 3 and Figure 4 show the ^1^H and ^13^C NMR spectra of the Lys-His-Arg dendrimer. Using ^1^H NMR spectra, the integral values of the peaks were calculated to estimate the contribution of protons of different groups to the signal.

### 2.2. Lys-Arg-His

On the proton NMR spectrum in Figure 1, we can see four regions where signals are observed. The signals in the aromatic region 8.30–7.00 ppm obviously refer to protons in the imidazole rings of the histidine residues. The signals in the region from 4.63 to 3.90 ppm belong to CH groups. The peaks in the range 3.30–2.90 ppm refer to the CH_2_ groups connected with nitrogen atoms or imidazole ring. The group of overlapping signals at 1.90–1.10 ppm corresponds to the CH_2_ groups of the aliphatic part of the dendrimer.

The ^13^C NMR spectrum of the Lys-Arg-His dendrimer is presented in Figure 2. The signals in the range of 178–170 ppm belong to carbons in carboxyl groups (symbols *a*, *d*, *j*, *l*, *n* in Figure 2). The peak at 156.70 ppm refers to quaternary carbon in guanidine groups of the arginine residues. The signals at 134.67, 130.30 and 117.24 ppm are attributed to carbons in the imidazole rings (symbols *w*, *u*, *v*, respectively, in Figure 1 and Figure 2). The signals in the range of 54–52.85 ppm belong to the carbon atoms in the CH groups. The peaks at 49.47 (symbols *z*, Figure 1 and Figure 2) and 39.07 ppm (symbols *i* and *s*, Figure 1 and Figure 2) refer to the CH_2_ groups adjacent to nitrogen atoms. The signals from the carbon atoms of the dendrimer’s aliphatic part are located in the region 30.65–16.80 ppm. 

The 2D ^1^H-^13^C HMBC and HSQC spectra of this dendrimer are presented in Appendix A, respectively, in Appendix A. Here, we present only the main results. It was found that the peaks at 8.10 and 7.10 ppm belong to the protons at the C_2_ carbon (symbol *w*, Figure 1) and at the C_4_ carbons (symbol *v*, Figure 1) of imidazole rings, respectively. We have the corresponding ^1^H-^13^C HSQC cross-peaks for the carbon atoms *w* (8.13, 134.92) and *v* (7.11, 117.23) (Appendix A). The cross-peaks (8.12, 117.11), (8.12, 130.37), (7.11, 130.40), and (7.10, 134.83) in the ^1^H-^13^C HMBC indicate a connection between the hydrogen and carbon atoms inside the imidazole ring In addition, the chemical shifts of carbon atoms in imidazole rings 134.67 (symbol *w*), 130.30 (symbol *v*) and 117.24 (symbol *u*) ppm are similar to the chemical shifts of these atoms in L-histidine: 138.87, 134.42 and 119.54 ppm, respectively.

The ^1^H-^13^C HMBC cross-peaks (3.11, 117.18), (3.11, 130.28) (Appendix A) confirm an interaction between carbons in the imidazole ring, with a proton at 3.11 ppm (symbol *t,* Figure 1). Then, according to the ^1^H-^13^C HSQC cross-peak (3.12, 27.52) the signal at 27.45 ppm refers to the CH_2_ groups connected to the imidazole rings (symbol *t*, Figure 1 and Figure 2) in histidine residues.

Cross peaks (3.15, 40.55) on ^1^H-^13^C HSQC and (3.14, 24.99), (3.14, 28.33) and (3.14, 156.65) on the ^1^H-^13^C HMBC spectra confirm that the carbons with the chemical shifts at 40.55 (symbol *z*, Figure 2), 24.99 (symbol *y*, Figure 2), 28.33 (symbol *x,* Figure 2), and 156.65 ppm (symbol *u’*, Figure 2) belong to arginine residues.

Further, based on the identified peaks we assigned the rest of the signals (see Appendix A). In addition, we clarified that the protons of the CH_2_ groups marked by symbols *i*, *t*, *z* contribute to signal at 3.13 ppm (Figure 1). The signal at 2.97 ppm is attributed to protons of the CH_2_ groups (symbol *s*, Figure 1) adjacent to N-atoms in terminal lysine segments.

### 2.3. Lys-His-Arg

The Lys-His-Arg dendrimer differs from the previous one in the order of insertion of histidine and arginine moieties, so we can see from Figure 3 and Figure 4 that the proton and carbon spectra are very similar to those for the Lys-Arg-His dendrimer. However, some differences should be noted. On the ^1^H spectrum (Figure 3), three peaks can be clearly observed in the region of 3.27–2.85 ppm. According to the ^1^H-^13^C HSQC spectrum (Figure 5 and Appendix A) and the calculated integral values, the protons CH_2_-(N) groups of the inner Lys (symbol *i*, Figure 3) and side Arg segments (symbol *z*, Figure 3) make the main contribution to the peak at 3.11 ppm. The contribution from the protons of the CH_2_ groups connected to imidazole rings in the side histidine segments predominates (symbol *t*, Figure 3) at 3.01 ppm. The peak at 2.96 ppm is attributed to the protons of the CH_2_-(N) groups of the terminal lysine segments (symbol *s*, Figure 4). In the case of Lys-Arg-His, only two peaks are observed in this region: a peak of the CH_2_-(N) groups of the terminal lysine segments (symbol *s*, Figure 1) at 2.97 ppm, and a common peak for other considered CH_2_ groups (symbols *i*, *z* and *t*, Figure 1) at 3.13 ppm. 

Detailed analyses of the ^1^H-^13^C HMBC and HSQC spectra are shown in Appendix A, respectively, in Appendix A.

In the Lys-His-Arg dendrimer, the signals from the protons of the imidazole ring (symbols *w* and *v* in Figure 3) shifted towards upfield compared to the Lys-Arg-His dendrimer. The difference is about 0.1 ppm. At the same time, the ^13^C spectrum (Figure 4) shows a slight increase in the chemical shifts of the carbon atoms of the imidazole ring *w* and *u*, as well as the carbon atoms *t* of the CH_2_ groups of the histidine segments. It is quite possible that the imidazole rings are geometrically close to each other and a pairing effect occurs (Figure 6) [59,60].

Before further analysis, we must become acquainted with the terminology, which we use in this study. As mentioned above, the peaks in the range of 3.13–2.90 ppm correspond to the signals of the CH_2_ groups connected to nitrogen atoms: “inner” groups in the inner lysine (symbol *i*) segments, “side” groups in the side arginine (symbol *z*), and histidine (symbol *t*, bonded to imidazole ring) segments and “terminal” groups in the terminal lysine (symbol *s*) segments. Note that we use these groups to study orientational mobility in peptide dendrimers, which will be discussed below.

In order to confirm the assumption about the presence of paring between the imidazole rings of neighboring histidine residues in the Lys-His-Arg dendrimer, let us consider for comparison the ^1^H spectra of the studied dendrimers and the Lys-2Arg and Lys-2His dendrimers in the range of 3.13–2.90 ppm in Figure 7. As can be seen, the shape and position of the peaks for the dendrimers Lys-Arg-His and Lys-2Arg are similar and two main peaks are observed at 3.15–3.05 ppm (inner and side groups) and 2.95–2.90 (terminal groups). In the case of the Lys-His-Arg dendrimer, splitting of the peak for inner and side groups occurs, which leads to the appearance of an additional peak at about 3.0 ppm. A similar situation is observed for the Lys-2His dendrimer. We believe that such a change in the spectra of dendrimers is also associated with the presence/absence of the pairing effect between the imidazole rings. This assumption can be confirmed using the spectrum of the Lys-2His dendrimer at low pH 1.1 (Lys-2Hisp) in which the imidazole rings are charged. At the same time, according to the results of atomistic MD simulation of the Lys-2His and Lys-2Hisp dendrimers, a pairing effect between imidazole rings is observed at a distance of 0.4 nm for Lys-2His, and is absent for Lys-2Hisp [61]. Figure 7 shows that in the spectrum of Lys-2Hisp there is practically no additional peak at 3.0 ppm.

Thus, we conclude that in the case of the Lys-Arg-His dendrimer, the pairing effect is absent, while in the case of the Lys-His-Arg dendrimer, the pairing effect is observed. It is possible that this is the effect of the formation of a collapsed conformation of the Lys-2His dendrimer, as well as the Lys-His-Arg dendrimer (which is presented below using NMR relaxation data).

### 2.4. Local Orientational Mobility

To study the local orientational mobility of the Lys-His-Arg and Lys-Arg-His dendrimers, we used the signals from the inner, side, and terminal groups (Figure 1, Figure 3 and Figure 7). According to the spectral analysis, we observed a resolved signal from the protons of the terminal groups and average signals from the inner and side groups, since they overlap. Among the dendrimers under consideration, Lys-2Arg is an exception, since there is a separate peak for inner groups [48,51,62]. This was taken into account when considering the temperature dependences of the spin-lattice relaxation rate 1/*T*_1H_ obtained (Figure 8).

In the framework of the dipole–dipole relaxation mechanism of ^1^H nuclei (protons), the 1/*T*_1H_ function can be written as [63,64,65,66,67]: (1)$$1/T_{1H}=A_{0}\left(J(\omega_{H},\tau_{i})+4J(2\omega_{H},\tau_{i})\right)$$
where *ω_H_* is the cyclic resonance frequency (2π*f*_0_) for ^1^H nuclei; *A*_0_ is a constant that does not depend on temperature and frequency; and *J* is the spectral density which corresponds to Fourier transform from *P*_2_ orientational autocorrelation functions averaged over groups contributing to a corresponding peak. In the general case, the spectral density function for ^1^H nuclei has the form:(2)$$J(n\omega_{H},\tau_{i})=\sum_{i}\frac{C_{i}\tau_{i}}{1+{\left(\tau_{i}n\omega_{H}\right)}^{2}}$$
where *τ_i_* and *C_i_* are *i*th correlation times and their contribution to *J*, respectively, and *n* = 1, 2. The correlation time is determined by Arrhenius dependence
(3)$$\tau =\tau_{0}\exp\left(\frac{E_{a}}{k_{b}T}\right)$$
where *E_a_* is the activation energy for the chosen group, and T and *k_b_* are temperature and Boltzmann constant, respectively. The theory of orientational mobility in dendrimer [68,69,70,71,72,73,74,75,76,77] predicts that the main contribution to NMR relaxation is provided by two processes with different correlation times: (i) rotation dendrimer as a whole and branch (or subbranch) reorientation. 

The local orientational mobility of groups in the dendrimer is determined by the position of the 1/*T*_1H_ maximum. Moreover, the more to the left of the maximum (i.e., shifted towards high temperatures), the slower the group mobility. In dendrimer macromolecules, the inner groups are the slowest (Figure 8a). The most mobile are the terminal groups, for which an exponential growth is observed in the 1/*T*_1H_ dependence, and the maximum is not reached due to the limited experimental temperature range (the freezing of the solvent) (Figure 8c). The mobility of side groups depends on the structure of the side segment and the position of the observed group in it. For example, in the case of the Lys-2Lys dendrimer, the mobility of the side groups is the same as the mobility of the terminal groups [51]. However, in Lys-2His, the mobility of the side groups coincides with the mobility of the inner groups [48].

As shown in [48], the position of the maximum of the 1/*T*_1H_ temperature dependence for inner groups shifts to the left (towards high temperatures) due to the transition from the swollen conformation to the collapsed one. Thus, if the structures of the dendrimers are similar, then the criterion for contraction of the dendrimer can be the shift of the maximum of the 1/*T*_1H_ dependence, to which the inner groups contribute, towards high temperatures. Figure 8a,b show the 1/*T*_1H_ dependences for the inner groups of Lys-2Arg (swollen conformation) and Lys-2His (collapsed conformation), which were obtained earlier and used as references.

Let us now proceed to analyze the relaxation data for the Lys-Arg-His and Lys-His-Arg dendrimers. As can be seen from Figure 8a, the 1/*T*_1H_ dependence for the side and inner groups of the Lys-Arg-His dendrimer practically coincides with the 1/*T*_1H_ curve for the inner groups of the Lys-2Arg dendrimer. This means that the mobility of the inner groups of Lys-Arg-His indicates the swollen conformation of the dendrimer. However, it should be noted that in the Lys-Arg-His dendrimer, the side arginine groups that contribute to this dependence have a higher mobility than the inner groups and side groups in histidine residues. For illustration, in Figure 8a, the 1/*T*_1H_ curve for the side and inner groups of the Lys-2Arg dendrimer is shown (solid red squares), the maximum for which is observed much more to the right than for the other groups. It can be expected that, in the case of Lys-Arg-His, the side arginine groups will also have a similar mobility, since its contribution is 40%. If it were possible to separate the signal only from the inner groups of the Lys-Arg-His dendrimer, then, obviously, the position of the 1/*T*_1H_ maximum would be to the left of the same maximum for the inner groups in Lys-2Arg. Thus, it can be concluded that the mobility of the inner groups in Lys-Arg-His is slower than in the Lys-2Arg dendrimer.

In the case of the Lys-His-Arg dendrimer, the 1/*T*_1H_ maximum is shifted by 10 K to the region of high temperatures compared to the similar curve for Lys-Arg-His. Therefore, taking into account the contribution of the side arginine groups, we can expect that the shift of the maximum will be even more significant and close to the 1/*T*_1H_ dependence for the Lys-2His dendrimer (in the collapsed conformation). Such a slowdown in the mobility of the inner groups indicates that the conformation of the Lys-His-Arg dendrimer is close to the collapsed one.

Analogous conclusions can be drawn from the 1/*T*_1H_ temperature dependence for the terminal groups (Figure 8c). The mobility of the terminal groups of the Lys-Arg-His dendrimer is similar to that of the Lys-2Arg dendrimer. At the same time, the 1/*T*_1H_ dependencies for the terminal groups of Lys-Arg-His and Lys-2His have a similar behavior.

Thus, according to the NMR relaxation data, it can be argued that, despite the same chemical composition, the Lys-Arg-His and Lys-His-Arg dendrimers have different conformational structures. The global conformation of the Lys-Arg-His dendrimer is close to collapsed, and the conformation of the Lys-His-Arg dendrimer is more swollen.

## 3. Conclusions

In our recent work [48], it was found that a peptide dendrimer with double histidine insertions changes its global conformation, from swollen at low pH (Lys-2Hisp) to collapsed at normal pH (Lys-2His). However, the biomedical usage of the Lys-2His dendrimer in a collapsed conformation can be problematic due to its tendency to aggregate.

This work is a continuation of our previous study in [48]. Here, we present the synthesis of new peptide dendrimers with the same amino acid composition, but different amino acid sequences (Arg-His or His-Arg), in insertions between lysine branching points. These dendrimers were experimentally investigated using NMR spectroscopy and NMR relaxation methods. The main idea for this research was to obtain a dendrimer macromolecule that retains the properties of size change due to the recharging of imidazole rings, but prevents the problem of aggregation of dendrimers with each other due to the insertion of charged guanidine groups in arginine residues.

Despite the same amino acid composition, the Lys-Arg-His and Lys-His-Arg dendrimers have different structural properties. In the case of the Lys-Arg-His dendrimer, the pairing effect of imidazole rings of the histidine residues does not appear in the NMR spectra, and the NMR relaxation behavior indicates a swollen conformation of the macromolecule, similar to that of the Lys-2Arg dendrimer. The opposite situation is observed for the Lys-His-Arg dendrimer: the shape and position of the peaks of the side histidine groups are similar to the corresponding peaks of the Lys-2His dendrimer, in which the paring effect is observed. Moreover, according to the NMR relaxation data, the conformation of the Lys-His-Arg dendrimer becomes close to the collapsed, as in the case of the Lys-2His dendrimer. 

Thus, we can make a conclusion that the Lys-His-Arg dendrimer is the most suitable candidate for use as a pH sensitive nanocontainer in biomedical applications.

## 4. Materials and Methods

### 4.1. Synthesis of Lys-His-Arg and Lys-Arg-His

Boc-amino acids were purchased from Bachem Holding (Torrance, CA, USA) and Iris Biotech GMBH (Marktredwitz, Germany); p-мethylbenzhydrylamine resin (MBHA-resin) was supplied by Bachem Holding (Torrance, CA, USA); trifluoromethanesulfonic acid (TFMSA), diisopropylcarbodiimide (DIC), 1-hydroxybenzotriazole (HOBt), thioanisole, ethanedithiol, and other reagents were purchased from Sigma-Aldrich (Munich, Germany). Triethylamine, dichloromethane (DCM) and dimethylformamide (DMF) were purchased from Vecton Ltd. (St. Petersburg, Russia). Trifluoroacetic acid (TFA) was purchased from Panreac (Barcelona, Spain) and distilled before application. All solvents were purified and distilled using standard procedures.

Lysine-based dendrimers with dipeptide insertions His-Arg and Arg-His between the branching points were obtained by a step-by-step BOC solid-phase peptide synthesis method (SPPS) [51]. The synthesis was carried out manually using 0.1 g MBHA-resin (amino group content was 0.72 mmol/g). The synthesis of the dendrimer molecules includes: (1) the introduction of L-alanine to initiate the synthesis; (2) branching using the trifunctional amino acid Boc-Lys(Boc); and (3) the formation of inner layers by insertion of histidine (His) and arginine (Arg) amino acid residues.

However, several points should be noted regarding the procedure for the synthesis of these dendrimers in order to obtain high quality products. Since the number of amino acids increased exponentially with each addition of Boc-Lys(Boc), the reaction was carefully controlled by test to avoid errors in the synthesis. During the formation of the first generation layers of macromolecules, we blocked unreacted peptide chains with an acetic anhydride/dichloromethane (1:1 *v*/*v*) solution for 30 min before washing the MBHA-resin with dimethylformamide (DMF), methanol, and dichloromethane (DCM) (twice each). When binding was difficult, especially on the terminal layers, N-methylpyrrolidone (NMP) was added to DMF. Probably some difficulties were caused by the presence of massive protective side groups: a p-benzyloxymethyl group (Bom) on histidine and a mesitylene-2-sulfonyl group (Mts) on arginine.

In the case of the Lys-His-Arg dendrimer, the average time of one acylation was increased to 10–12 h instead of 4–6 h, especially with increasing generations. In addition, the introduction of histidine residues in the first layer led to certain difficulties in the synthesis, compared with the Lys-Arg-His dendrimer, in which the arginine layer was the starting one. The first layer of histidine not only increased the binding time and the excess of reagents up to 6 equivalents, but also caused problems with the subsequent insertion of arginine residues. Furthermore, for complete conversion during the formation of the second generation layer of the Lys-His-Arg dendrimer, in addition to increasing the time and quantity of reagents, it was necessary to add 4-N,N-dimethylaminopyridine (DMAP) (0.1 eq.) as a catalyst.

In the case of the Lys-Arg-His dendrimer, practically no reacylation of the branching lysine layer was required, except for the terminal one; the other stages took twice as much time and twice as many reagents. Despite this, the amino groups in each generation remained available for complete conversion.

### 4.2. NMR Experiments

For the NMR study, the samples of the Lys-Arg-His and Lys-His-Arg dendrimers were dissolved in 0.158 M NaCl D_2_O at a concentration of about 1.54 g/dl and 1.59 g/dl, respectively.

NMR measurements were performed on a Bruker Avance III 500 MHz spectrometer (Karlsruhe, Germany) equipped with a standard 5 mm BBFO direct observation probe and a Great 1/60A gradient system with a MIC S2 Diff/30 diffusive probe with a ^1^H convolution compensation coil (EVT). One- and two-dimensional NMR spectra for peak assignments were recorded at 298 K. All spectra were obtained using standard pulse sequences. The ^1^H spin-lattice relaxation times, *T*_1H,_ were acquired with an “inversion-recovery” pulse sequence, modified by the destructive gradient pulses at the beginning of the sequence (“spoiler-recovery” sequence) [78]. NMR relaxation experiments were carried out at the temperature range 283–343 K.

## Figures and Tables

**Figure 1 ijms-24-00949-f001:**
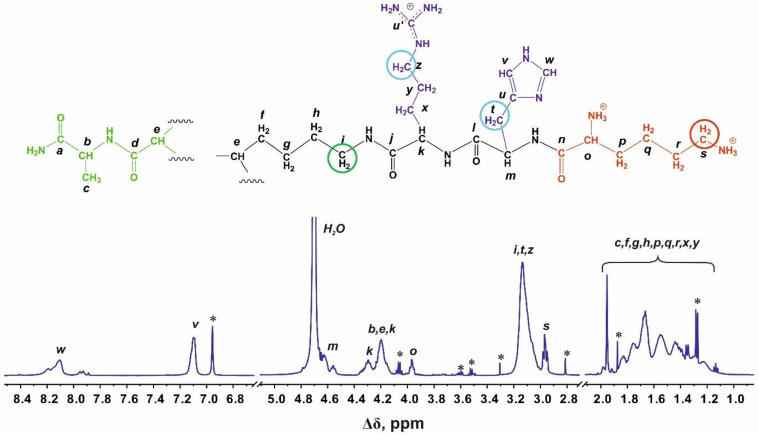
^1^H NMR spectrum of the Lys-Arg-His dendrimer at 298 K. The dendrimer contains three types of methylene groups connected with nitrogen atoms (or imidazole rings in His residues), which have been used in NMR relaxation study: inner groups (green open circle), side groups (blue circles) and terminal groups (red circle). The peaks from small molecular weight impurities are marked by asterix (*).

**Figure 2 ijms-24-00949-f002:**
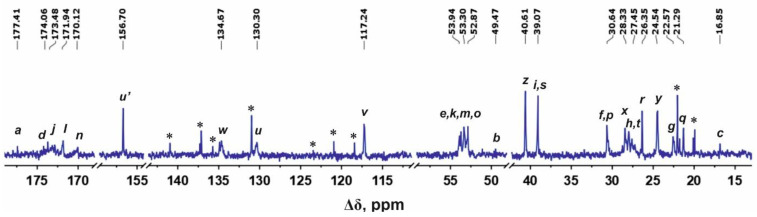
^13^C NMR spectrum of the Lys-Arg-His dendrimer at 298 K. The peaks from small molecular weight impurities are marked by asterix (*).

**Figure 3 ijms-24-00949-f003:**
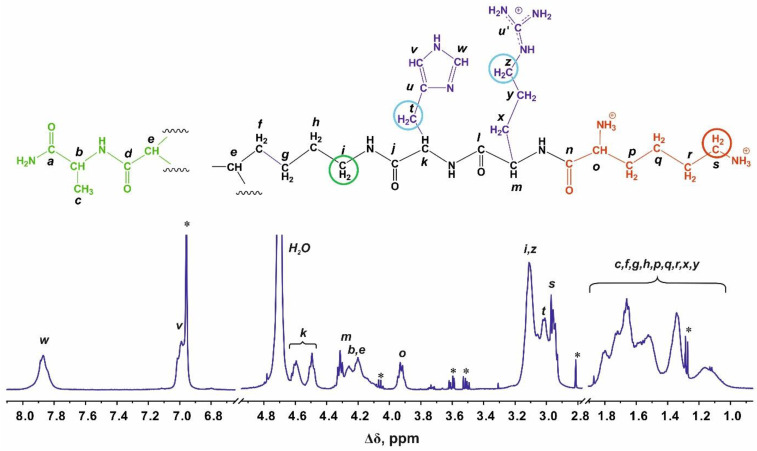
^1^H NMR spectrum of the Lys-His-Arg dendrimer at 298 K. Three types of methylene groups connected with nitrogen atoms (or imidazole rings in His residues), which have been used in NMR relaxation study: inner groups (green open circle), side groups (blue circles) and terminal groups (red circle). The peaks from small molecular weight impurities are marked by asterix (*).

**Figure 4 ijms-24-00949-f004:**
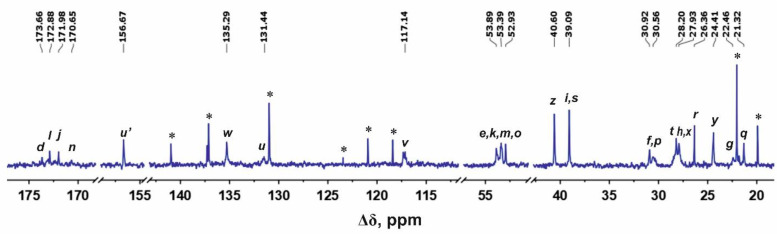
^13^C NMR spectrum of the Lys-His-Arg dendrimer at 298 K. The peaks from small molecular weight impurities are marked by asterix (*).

**Figure 5 ijms-24-00949-f005:**
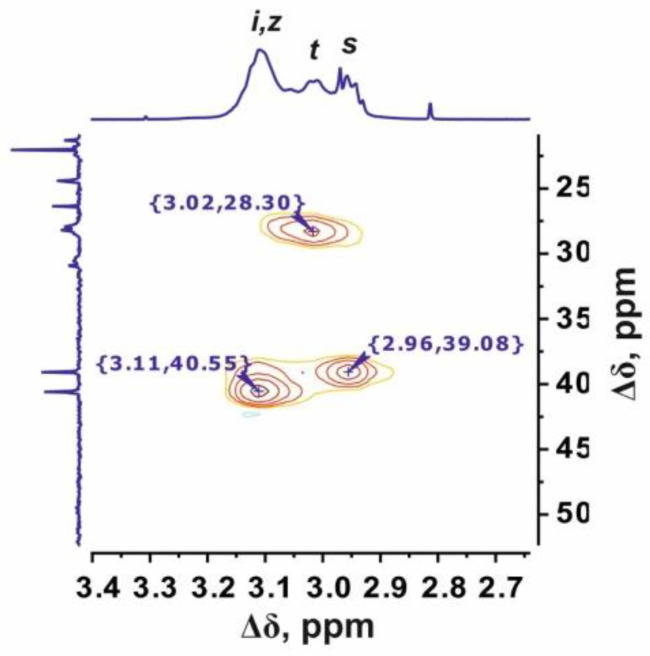
^1^H-^13^C HSQC NMR spectrum of the Lys-His-Arg dendrimer in the range of 3.40–2.65 ppm at 298 K.

**Figure 6 ijms-24-00949-f006:**
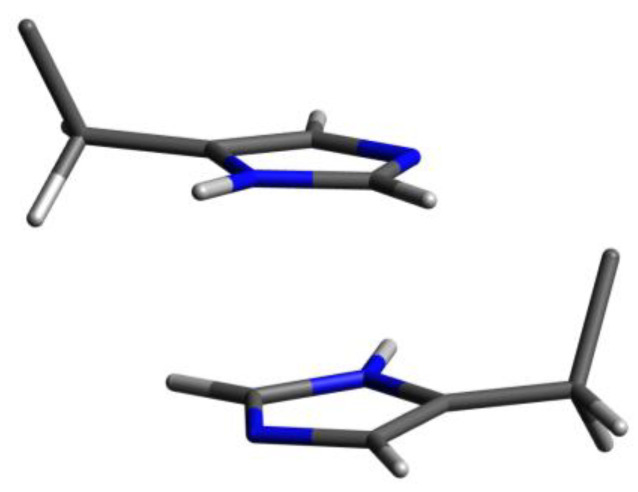
The schematic illustration of a pairing between imidazole rings. Carbon is grey; nitrogen is blue; hydrogen is white.

**Figure 7 ijms-24-00949-f007:**
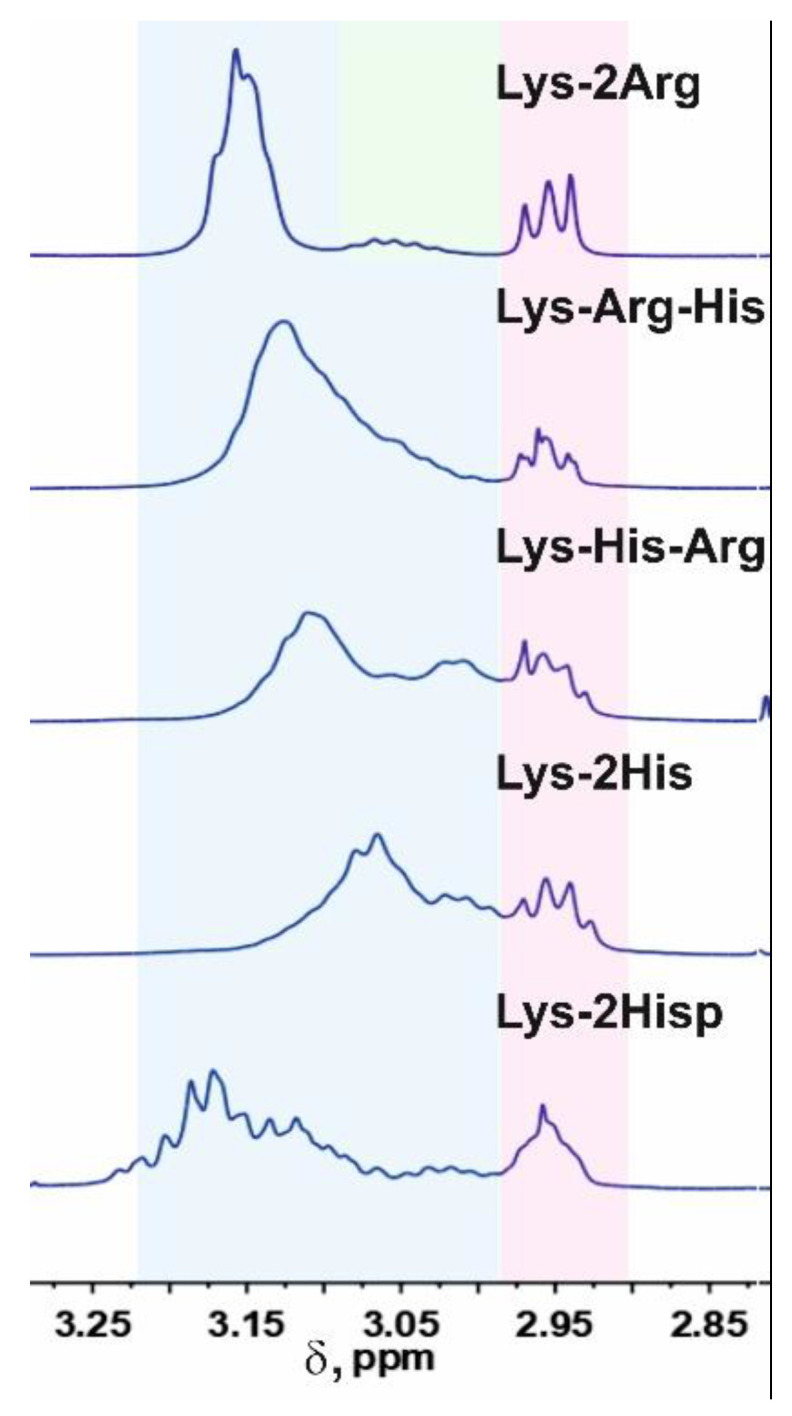
Comparison of ^1^H NMR spectra of different peptide dendrimers in the region of signals from inner CH_2_-(N) (green), side and inner (blue) and terminal (red) groups at a temperature of 298 K.

**Figure 8 ijms-24-00949-f008:**
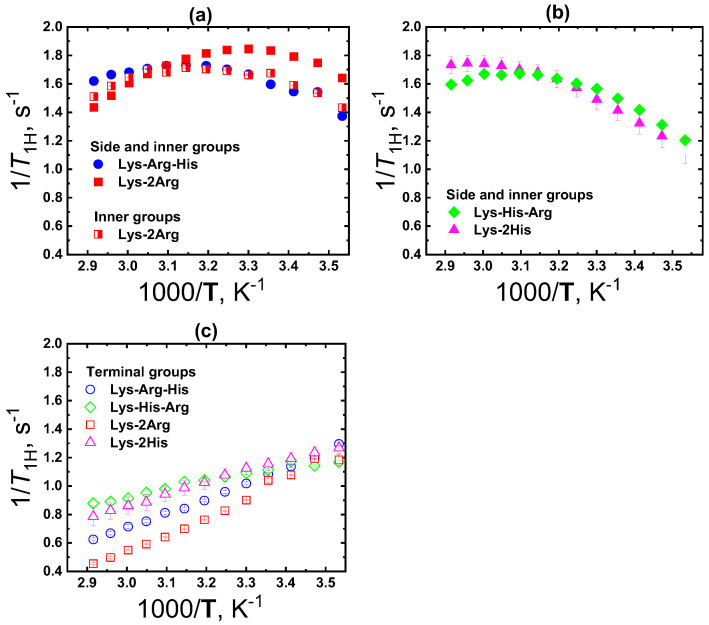
Temperature dependences of the spin−lattice relaxation rate, 1/T_1H_: of (**a**) side and inner CH_2_ groups of the Lys-Arg-His and Lys-2Arg dendrimers and inner groups of Lys-2Arg; (**b**) side and inner CH_2_ groups of the Lys-His-Arg and Lys-2His dendrimers; (**c**) terminal groups of Lys-Arg-His, Lys-His-Arg, Lys-2Arg and Lys-2His dendrimers.

## Data Availability

Not applicable.

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
