# Peer review of "NMR Studies of Two Lysine Based Dendrimers with Insertion of Similar Histidine-Arginine and Arginine-Histidine Spacers Having Different Properties for Application in Drug Delivery"

_ijms, 2023, doi:10.3390/ijms24020949_

Round 1
Reviewer 1 Report
The article is well written, I only have a few comments
1- I do not see the need to use 20 references to stress that "nanosized macromolecules are widely used in biomedicine as carriers in gene and drug delivery" in line 33.
2- In the two shown 13C NMR spectra the peak at 156.7 ppm is a sharp peak with high intensity, I think that is too sharp for a quaternary positively charged carbon surrounded by three nitrogen atoms! I would expect the peak for that atom to be similar to the signals w or u or even broader or not even to detect it. I have looked at the reference used (53) which is a publication of the same group, I would have the same issue with that reference too.
As seen from the 13C spectra most of the dendrimer peaks have low intensity and are broad peaks, so I do not think the peak at 156.7 and the peaks in the next point are for moieties of the macromolecule dendrimer, but rather for small molecular weight impurities.
see for example (N-Boc-N'-nitro-L-arginine) which has a 1H peak at about 7 ppm and 13C peak for the Boc group at 156.7 ppm.
3- The are a few unassigned peaks in the 1H and 13C NMR spectrum, such as 4.05, 6.95 ppm in Fig 1 & Fig 2, also peaks at 3.5, 3.5 ppm in Fig 2. In the 13C there are peaks 131, 136, 141 ppm, and many others!
4- No results were presented nor discussed from the HMBC and only one point from the HSQC.
Author Response
We would like to thank the Reviewer for his/her work. We hope that in its present form the manuscript will be suitable for publication in International Journal of Molecular Science. Please, find the attached file with detailed answers to the reviewer's questions.

Reviewer 2 Report
In this paper, Nadezhda et al. continue their precious work and studied two lysine-based peptide dendrimers with Lys-His-Arg and Lys-Arg-His. In addition, the authors investigated the conformation of these two lysine-based peptide dendrimer macromolecules using NMR spectroscopy. It is interesting to find that the Lys-Arg-His and Lys-His-Arg dendrimers, with the same amino acid composition, yet exhibit different structural properties. Overall, this work has performed very well in detailed experimental studies and mechanistic analysis and has important implications for the study of pH-sensitive peptide supramolecular materials and peptide-based drug nanocarriers. I would like to recommend it for publication in IJMS after the following points can be addressed.
Minor point:
1. Authors should include some simple structural schemes of these two different conformations of lysine-based peptide dendrimers (collapsed conformation and swollen conformation) in the manuscript for better illustration and explanation.
2. Similarly, it is better to add one figure to illustrate the pairing effect between imidazole rings in the Lys-His-Arg dendrimer.
3. Error bar should be indicated in Figure 7.
4. Reference selection is good in the manuscript. It is recommended to add recent articles on design strategies and applications of histidine-functionalized peptide-based supramolecular materials. (e.g., 10.1016/j.mattod.2022.08.011; 10.1021/jacs.1c11750; 10.1002/anie.202105830).
Author Response

(The authors gave the same response as above.)

Round 2
Reviewer 1 Report
Not much to add,